# A pilot study to evaluate the magnitude of association of the use of electronic personal health records with patient activation and empowerment in HIV-infected veterans

Pierre-Cédric B. Crouch[1], Carol Dawson Rose[1], Mallory Johnson[2] and Susan L. Janson[1]

[1] Department of Community Health Systems, University of California, San Francisco, CA, USA
[2] Center for AIDS Prevention Studies, University of California, San Francisco, CA, USA

Corresponding author
Pierre-Cédric B. Crouch,
pcrouch@sfaf.org

## ABSTRACT

The HITECH Act signed into law in 2009 requires hospitals to provide patients with electronic access to their health information through an electronic personal health record (ePHR) in order to receive Medicare/Medicaid incentive payments. Little is known about who uses these systems or the impact these systems will have on patient outcomes in HIV care. The health care empowerment model provides rationale for the hypothesis that knowledge from an electronic personal health record can lead to greater patient empowerment resulting in improved outcomes. The objective was to determine the patient characteristics and patient activation, empowerment, satisfaction, knowledge of their CD4, Viral Loads, and antiretroviral medication, and medication adherence outcomes associated with electronic personal health record use in Veterans living with HIV at the San Francisco VA Medical Center. The participants included HIV-Infected Veterans receiving care in a low volume HIV-clinic at the San Francisco VA Medical Center, divided into two groups of users and non-users of electronic personal health records. The research was conducted using in-person surveys either online or on paper and data abstraction from medical records for current anti-retroviral therapy (ART), CD4 count, and plasma HIV-1 viral load. The measures included the Patient Activation Measure, Health Care Empowerment Inventory, ART adherence, provider satisfaction, current CD4 count, current plasma viral load, knowledge of current ART, knowledge of CD4 counts, and knowledge of viral load. In all, 40 participants were recruited. The use of electronic personal health records was associated with significantly higher levels of patient activation and levels of patient satisfaction for getting timely appointments, care, and information. ePHR was also associated with greater proportions of undetectable plasma HIV-1 viral loads, of knowledge of current CD4 count, and of knowledge of current viral load. The two groups differed by race and computer access. There was no difference in the current CD4, provider satisfaction, Health Care Empowerment Inventory score, satisfaction with provider-patient communication, satisfaction with courteous and helpful staff, knowledge of ART, or ART adherence. The use of electronic personal health records is associated with positive clinical and behavioral characteristics. The

use of these systems may play a role in improving the health of people with HIV. Larger studies are needed to further evaluate these associations.

## BACKGROUND

Electronic Personal Health Records (ePHR) are systems that allow patients to access their health information through patient portals linked to their health records. With the passage of the Health Information Technology for Economic and Clinical Health Act (HITECH) in 2009, which provides increased Medicare and Medicaid reimbursement to providers who have adopted Electronic Health Records (EHR) and ePHR, these systems will become commonplace. The law states that these systems must provide a "meaningful use" in order to qualify for incentive payments to implement these systems. There are three stages with specific implementation criteria that are used to define meaningful use. Each stage builds upon the other and sets functional requirements in an EHR that need to be met in order to meet the definition of meaningful use. Stage 2 is currently underway in 2014. To qualify for the incentive payments under meaningful use stage 2, eligible hospitals will need to demonstrate that: (1) 50% of unique patients were provided with timely online access to their health information, (2) 5% of unique patients view, download or transmit their health information to a third party, and (3) 5% of unique patients send a secure message (*Centers for Medicare and Medicaid Services, 2013*).

Little is known about the potential impact of electronic access to personal health information on one's sense of health care empowerment and activation to participate in healthcare. However, among patients with HIV, those who are more engaged and empowered in their health care have higher t-cell counts, lower viral loads, greater medication adherence, and participate in less risky behaviors (*Marks et al., 2005*; *Berg et al., 2005*; *Park et al., 2007*; *Metsch et al., 2008*; *Mugavero et al., 2010*; *Marshall et al., 2013*). Access to personal health information in ePHR systems may help patients with HIV feel more empowered and activated in their health care by providing the knowledge needed to care for themselves. Therefore, the purpose of this study was to explore the patient characteristics, healthcare empowerment, patient activation, and satisfaction associated with electronic personal health record use in Veterans living with HIV under care at the San Francisco VA Medical Center.

The Department of Veteran Affairs has been a leader of ePHR development with its system named My HealtheVet, which launched in 2003 (*United States Department of Veterans Affairs, 2013*). The My HealtheVet ePHR is a tethered system that allows all patients at all of the VAMC to have full access to their lab results, progress notes, and medication refills. It also gives patients the ability to send secure messages to their

providers at http://www.myhealth.va.gov. Patients are required to have their identity authenticated in person before obtaining access.

## THEORY

The theory used to guide this research is the Health Care Empowerment Model developed by psychologist Dr. Mallory Johnson in 2011 *Johnson et al. (2012)*. This model provides a framework that "would enable organization, investigation, and intervention upon factors that contribute to optimal health outcomes" (p. 265). The model of empowerment is comprised of five concepts: engaged, informed, collaborative, committed, and tolerant of uncertainty. These five concepts make up the construct of Health Care Empowerment.

The Health Care Empowerment Model can guide research on the value of patient access to ePHR because these systems provide a source of information that can directly impact the patient's engagement in personal health care. The information from the ePHR may result in increased access to care and promote active participation, which may influence engagement in a patient's own health care. The variables collected in this study were selected in reflection of this model and to describe the characteristics of these patients.

## METHODS

### Study design

A cross-sectional observational survey pilot study and chart data abstraction was conducted to describe characteristics of HIV-infected users and non-users of electronic personal health records. The Committees on Human Research at the University of California, San Francisco approved this study (CHR# 12-08729), which is the IRB of record for the San Francisco VA Medical Center.

### Participants

From September to December 2013, participants were recruited through distribution of flyers describing study eligibility and contact information, electronic health record screening to identify eligible participants, and through healthcare provider referrals. The participants either contacted the researcher directly or they were approached in person while in the waiting room of the dedicated HIV clinic.

A non-probability quota sample was used to identify 20 participants who used My HealtheVet and 20 participants who did not use My HealtheVet. Each group was recruited simultaneously until each group was filled. The group was divided by ePHR use only and no other variables. A total sample size of 40 was used as the minimum required for a pilot study to assess associations (*Hertzog, 2008*). The inclusion criteria were self-reported HIV infection, ability to speak and understand English, age of 18 years old or older, status as a military Veteran receiving care at the (SFVAMC), use of the My HealtheVet ePHR system at least twice in the last year (for a subset of 20 participants), and non-use of My HealtheVet ePHR system since its release (for a subset of 20 participants). Participants who lacked the capacity to provide informed consent due to active psychosis, cognitive impairment, significant confusion or intoxication were excluded.

## Setting

The San Francisco VA Medical Center provides comprehensive outpatient HIV care to 550 Veterans living with HIV in the San Francisco Bay area through services in the Infectious Disease Practice. This clinic is a low volume clinic primarily treating Veterans with HIV and providing infectious disease consultations to non-HIV infected veterans.

## Study procedures

Participants were recruited either during or outside of clinic hours. After determining eligibility and obtaining written informed consent, participants were given the option, depending on their self-reported computer literacy level, to complete either an online questionnaire, paper questionnaire, or have the questionnaire administered to them by the researcher. All participants were informed that the research was optional, had no relationship to their care, and that they could skip any questions that they did not feel comfortable answering in the survey.

## Measures

Demographics and comorbidities were collected using a self-report checklist. Use of the My HealtheVet system was ascertained through self-report and verified in the EHR. Participants were considered users of the system if they had accessed it at least twice within the last 12 months, and had accessed at least one of the available features, such as accessing their health data, ordering medication refills, or sending a secure message to their provider. Access to a computer was measured through self-report. Veterans were considered to have access to a computer based on their self-reported access to computers at home, school, library, or any other source.

Patient activation was measured using the Patient Activation Measure (PAM-13). The PAM-13 is a 13 item questionnaire with Likert scale responses that assess patient knowledge, skill, and confidence for self-care (*Hibbard et al., 2005*). The activation score ranges from 0 to 100 and is derived from the raw scores based on Likert responses. There are four levels in the activation score used to identify each participant's stage of activation. Level 1 is scores less than 47, and reflects a participant who may not believe the patient role is important. Level 2 is scores ranging from 47.1 to 55.1, and indicates participants who lack the confidence and knowledge to take action. Level 3 is scores ranging from 55.2 to 67.0, and is for participants who are beginning to take action. Level 4 is scores ranging from 67.1 to 100, and indicates participants who are "staying the course" under stress. The PAM-13 is valid and reliable with a Cronbach's alpha of 0.91 (*Hibbard et al., 2004*) and a Rasch person statistic of 0.81 for the real and 0.85 for the model on which it was based (*Hibbard et al., 2005*).

Patient Empowerment was measured with the Health Care Empowerment Inventory (HCEI). The HCEI is an 8-item questionnaire with Likert scale responses used to assess five hypothesized interrelated facets of the Health Care Empowerment Model (*Johnson et al., 2012*). These include: informed, engaged, committed, collaborative, and tolerant of uncertainty. The measure provides two scores: a composite measure of informed, engaged,

committed, and collaborative (HCEI_ICCE) and a measure of tolerant of uncertainty (HCEI_Tol). The HCEI is reliable with a rho of 0.78 for HCEI_ICCE and a rho of 0.86 for HCEI_Tol. It also has construct validity demonstrated with factor analysis (*Johnson et al., 2012*).

Patient satisfaction was measured with the Consumer Assessment of Healthcare Providers and Systems (CAHPS) (version 2.0) (*Agency for Health Care Research and Quality, 2013*). This survey consists of 14 items, including a single-item global provider rating, and three composite scores measuring three concepts: courteous and helpful office staff, provider-patient communication, and getting timely appointments, care, and information. The single provider rating score is a 1 to 10 on provider satisfaction with a 10 indicating the "best possible provider." The composite satisfaction scores are the mean score of 4-item Likert response of never (1), sometimes (2), usually (3), and always (4) giving an individual item score out of 4. The concept of courteous and helpful office staff composite score is the mean response of two questions. The concept of provider-patient communication composite score is the mean response of six questions. The concept of getting timely appointments, care, and information composite score is the mean response of five questions. These composite scores have a Cronbach's alpha ranging from .58–.75 and construct validity confirmed with factor analysis (*Hargraves, Hays & Cleary, 2003*).

Medication adherence was measured using the Community Programs for Clinical Research 7-day adherence measure. The adherence scale uses the Likert responses: all (100%), most (80%), about half (50%), very few (20%), none (0%) to assess the self-reported percentage of antiretroviral therapy (ART) taken in the last 7 days. In one previously published study, patients who reported 100% adherence had higher CD4 counts and lower plasma HIV-1 RNA than those who reported less than 100% indicating criterion validity (*Mannheimer et al., 2002*).

Knowledge of CD4 counts, plasma viral loads, and ability to identify current ART were determined through survey questions. CD4 counts were considered correct if the participants were able to correctly identify their CD4 counts within the ranges of less than 200, 200–500, and over 500. The research team selected these ranges for their clinical significance to the clinicians and patients. Knowledge of plasma HIV-1 viral loads was considered correct if the participants were able to accurately identify their viral loads as being detectable versus undetectable (<40 copies/ml). Ability to identify ART was considered correct if the participants were able to provide a phonetic spelling of their ART regimen as determined by the investigator.

## Analysis

The two groups, My HealtheVet users and non-My HealtheVet users, were compared for differences using bivariate comparisons of the demographic data and characteristic variables. All analyses were conducted using STATA 13.0. *T*-tests were performed on normally distributed data. Wilcoxon-Mann–Whitney tests were performed on non-normally distributed data. Fisher exact tests and Chi-square tests were performed on proportional data.

## RESULTS

Of the 43 participants approached, 40 (93%) agreed to participate in the study. There was no missing data in the reported data. The final analyses consist of 20 My HealtheVet users and 20 non-users.

### Sample characteristics

Table 1 describes the demographic data for all the participants and the two comparison groups. The two groups differed by undetectable plasma HIV-1 RNA, with 19 (95%) of the My HealtheVet users having an undetectable viral load and 14 (70%) of the non-My HealtheVet users Fisher = .046. The two groups also differed by race with a Fisher= .01 and access to a computer with a Fisher= .046. There were no differences by age, education, gender (male, female, transgender), current CD4 count, or comorbidities (Diabetes, Hypertension, Depression, Anxiety, PTSD, Hepatitis B, and Hepatitis C). Table 2 describes the users of My HealtheVet who reported a mean satisfaction of 8.1 (SD = 2.46). The most frequently used service was to refill medications followed by reviewing lab results. The mean number of times My HealtheVet was accessed in the last year was 15.25 times (SD = 11.76).

### Outcome variables

Table 3 shows the results of the activation, empowerment, satisfaction, and medication adherence variables. The mean PAM-13 score was 67.99 (SD = 12.47), indicating a level 4 of activation, which constitutes "staying the course" under stress. The mean HECI_ICCE score was 17.35 (SD = 2.33) and the mean HCEI_Tol score was 15.93 (SD = 2.60). The mean provider satisfaction score was 9.22 (SD = 1.07). The composite satisfaction score for courteous and helpful office staff was 3.59 (SD = 0.55). The composite score for provider-patient communication was 3.69 (SD = 0.42). The composite score for getting timely appointments, care, and information was 2.86 (SD = 0.71). Twenty-five (63%) of the participants were able to correctly report their CD4 count. Twenty-seven (68%) were able to correctly report their HIV viral loads. Eighteen (45%) were able to correctly report their ART regimens. Thirty-seven (93%) reported full adherence to ART. The two groups differed by PAM-13 scores, with My HealtheVet users reporting a higher mean PAM-13 score of 72.5 (11.27) versus a mean of 63.49 (12.23) $z = 2.21$, $p = .03$. The My HealtheVet users reported greater satisfaction on getting timely appointments, care, and information with a mean of 3.1 (0.6) versus a mean of 2.63 (0.75) $z = 2.15$, $p = .03$ for the non-users. The My HealtheVet users were able to correctly identify their CD4 counts, with 16 (80%) versus 9 (45%) fisher = .048, as well as their viral loads, with 18 (68%) versus 9 (45%) fisher = .003. The groups did not differ by HECI_ICCE scores, HCEI_Tol scores, provider satisfaction scores, courteous and helpful office staff scores, provider-patient communication, identifying ART regimen, or reporting 100% adherence.

### Digital divide

The difference of use of ePHR by race and access to computers was suggestive of a digital divide that may have been present in this population, prompting a sub analysis to determine if access to a computer differed by race. The Fisher exact test equaled .171,

**Table 1** Sample characteristics and subsample comparisons.

| Variable | All n = 40 | My HealtheVet user n = 20 | My HealtheVet non-user n = 20 | Statistics |
|---|---|---|---|---|
| Age in years | 44.65(10.28) | 42.5 (10.82) | 46.8 (9.49) | $t = -1.34(38)$ $p = .19$ |
| CD4 cell/ml$^3$ | 580.73(254.68) | 515.15(263.94) | 646.3(233.28) | $z = -1.73$, $p = .08$[a] |
| On ART | 40(100%) | 20(100%) | 20(100%) | Fisher = 1.00 |
| Undetectable plasma HIV-1 RNA | 33(83%) | 19(95%) | 14(70%) | Fisher = .046 |
| Gender | | | | |
| Male | 39(98%) | 19(95%) | 20(100%) | |
| Female | 0 | 0 | 0 | |
| Transgender male to female | 1(2%) | 1(5%) | 0 | Fisher = 1.00 |
| Hispanic | | | | |
| Yes | 6(15%) | 3(15%) | 3(15%) | |
| No | 34(85%) | 17(85%) | 17(85%) | Fisher = 1.00 |
| Race | | | | |
| Caucasian | 31(77.5%) | 19(95%) | 12(60%) | |
| African American | 4(10%) | 0 | 4(20%) | |
| Native American | 2(5%) | 1(5%) | 1(5%) | |
| Other | 3(7.5%) | 0 | 3(15%) | Fisher = .01 |
| Education | | | | |
| Less than HS | 1(2.5%) | 0 | 1(5%) | |
| HS/GED | 8(20%) | 4(20%) | 4(20%) | |
| Some college | 7(17.5%) | 3(15%) | 4(20%) | |
| 2-year college | 8(20%) | 2(10%) | 6(30%) | |
| 4-year college | 10(25%) | 7(35%) | 3(15%) | |
| Masters degree | 4(10%) | 3(15%) | 1(5%) | |
| Doctoral degree | 2(5%) | 1(5%) | 1(5%) | Fisher = .45 |
| Diabetes | 9(22.5%) | 5(25%) | 4(20%) | Fisher = 1.00 |
| Depression | 11(27.5%) | 6(30%) | 5(25%) | chi$^2 = 0.07(1)$, $p = 0.8$ |
| Anxiety | 5(12.5%) | 2(10%) | 3(15%) | Fisher = 0.66 |
| PTSD | 2(5%) | 1(5%) | 1(5%) | Fisher = 1.00 |
| HTN | 24(60%) | 14(70%) | 10(50%) | chi$^2 = 1.67(1)$, $p = 0.2$ |
| HBV | 14(35%) | 4(20%) | 10(50%) | Fisher = 0.09 |
| HCV | 8(20%) | 4(20%) | 4(20%) | Fisher = 1.00 |
| Computer access | 33 | 19 | 14 | Fisher = .046 |

**Notes.**
[a] Non-parametric data.

indicating that access to a computer did not differ by race in this sample. This sub analysis showed that the difference in using My HealtheVet by race was not due to a lack of access to a computer. However, this finding does not exclude the possibility of other racial disparities that may influence the difference in ePHR use.

**Table 2 My HealtheVet user characteristics.**

| Variable | My HealtheVet user n = 20 |
|---|---|
| MHV satisfaction | 8.1(2.46) |
| MHV frequency in last year | 15.25(11.76) |
| Uses | |
|     Secure messaging | 9(45%) |
|     Review labs | 10(50%) |
|     Review notes | 6(30%) |
|     Review appointments | 8(40%) |
|     Refill medications | 18(90%) |

**Table 3 Activation, empowerment, satisfaction, and adherence outcomes.**

| Variable | All n = 40 | My HealtheVet user n = 20 | My HealtheVet non-user n = 20 | Statistics |
|---|---|---|---|---|
| PAM-13 | 67.99(12.47) | 72.5(11.27) | 63.49(12.23) | $z = 2.21, p = .03$[a] |
| HCEI_ICCE | 17.35(2.33) | 17.80(2.33) | 16.90(2.29) | $z = 1.23, p = .22$[a] |
| HCEI_TOL | 15.93(2.60) | 16.20(2.78) | 15.65(2.43) | $z = 0.63, p = .53$ |
| Provider satisfaction | 9.22(1.07) | 9.45(0.76) | 9.00(1.30) | $z = 0.88, p = .38$[a] |
| Courteous and helpful office staff | 3.59(0.55) | 3.6(0.60) | 3.58(0.52) | $z = 0.31, p = .76$[a] |
| Provider–patient communication | 3.69(0.42) | 3.76(0.33) | 3.62(0.49) | $z = 0.57, p = .57$[a] |
| Getting timely appointments, care, and information | 2.86(0.71) | 3.1(0.6) | 2.63(0.75) | $z = 2.15, p = .03$[a] |
| CD4 correct | 25(63%) | 16(80%) | 9(45%) | Fisher = .048 |
| VL correct | 27(68%) | 18(90%) | 9(45%) | Fisher = .003 |
| ART correct | 18(45%) | 11(55%) | 7(35%) | $chi^2 = 1.62(1), p = .2$ |
| Full adherence | 37 | 19 | 18 | Fisher = 1.00 |

**Notes.**

[a] Non-parametric data.

## Discussion

The use of My HealtheVet is associated with higher levels of activation, lower plasma HIV-1 RNA, greater ability to correctly identify CD4 counts and viral loads, and higher satisfaction in getting timely appointments, care, and information. The two groups also did not differ by adherence to ART. The adherence measure results lacked variability as many reported a high level of adherence, which makes it difficult make conclusions based on this one adherence measure.

The two groups differed by race and access to a computer. This is a significant finding, as patients without access to a computer could never potentially benefit from having access to their health information. This finding indicates the possibility of a digital divide present in the clinic. The *Oxford dictionary (2014)* defines a digital divide as "the gulf between those who have ready access to computers and the Internet, and those who do not." A digital divide present in the clinic could influence the characteristics of ePHR users. While this study was not designed to fully capture a digital divide, the sub analysis indicates that the

racial disparity in ePHR use is not associated with having a computer in order to access the ePHR but may be indicative of some other type of barrier.

Overall, we speculate that the use of electronic personal health records provides patients with more knowledge and opportunities to participate in their care than traditional visits only care. To our knowledge, this is the first study assessing activation and empowerment of electronic personal health record users in HIV care without any additional interventions. Three previously published studies of ePHR use among HIV infected patients addressed different goals and populations. Those findings indicated that healthy young Caucasian men typically use these systems and that they are largely viewed as useful tools in their care, but some participants expressed concerns about the accuracy of their health data in the ePHR (*Kahn et al., 2010*). Mental health and substance abuse issues did not seem to affect a participants' ability to access information online (*Hilton et al., 2012*). In addition, innovative methods of delivering ePHR though an Apple iPod device were accepted by participants and resulted in greater self-efficacy for self-care (*Luque et al., 2013*).

Studies of ePHR in other chronic diseases varied in their findings. The use of ePHR in the context of diabetes care reflected consistently positive findings in biological markers of diabetes control and quality of life. The frequency of ePHR use was found to have a great impact on the outcomes with more frequent users of ePHR having greater improvement in biological markers (*Smith et al., 2004*; *Hess et al., 2007*; *Zickmund et al., 2008*; *Fonda et al., 2009*; *Holbrook et al., 2009*).

Other conditions also benefited from ePHR. Patients with asthma (*Van der Meer et al., 2009*) or with hypertension (*Green et al., 2008*) were found to benefit from the use of ePHR with improved control of their illnesses *Wagner et al. (2012)* found no benefit of ePHR use in hypertension but a secondary analysis found that frequent users of ePHR had a significantly lower diastolic blood pressure. Patients undergoing *in vitro* fertilization (*Tuil et al., 2007*) or with multiple sclerosis (*Miller et al., 2012*) did not have any improvement in self-efficacy for self care from the use of ePHR. A study of an internal medicine practice demonstrated improved patient satisfaction of clinic communication and a greater likelihood of using ePHR to send messages about psychosocial issues and information-only messages (*Lin et al., 2005*).

## Limitations

The findings of our study should be interpreted with caution. This was a pilot study with a small sample of mostly male military Veterans. The pilot approach was used to explore the knowledge and activation characteristics of patients who use My HealtheVet. The study was not powered to find significant differences in some of the variables. The results demonstrate associations only and do not indicate a causal relationship between ePHR use and health behaviors. The study involved participants who had previously enrolled in the My HealtheVet system, which may demonstrate a pre-existing interest of this cohort in their own care. The study was based on self-reported behaviors of adherence and use of the My HealtheVet ePHR, which is subject to recall bias. The study sample was drawn from an older HIV-Infected and primarily male population receiving care in a highly skilled,

low volume setting where providers can spend more time with their patients, introducing attention as another source of potential bias.

Future research is needed to fully address the impact of ePHR in HIV care. A randomized controlled study is needed to determine if ePHR systems can independently cause changes in patient activation. Potential digital divides among populations may also play an important role in determining who is able to benefit from these electronic systems.

## ACKNOWLEDGEMENTS

I would like to thank the Veterans at the San Francisco VA who helped make this study a success and my dissertation committee for supporting me during this project.

**Note.** The views expressed here are those of the authors and not necessarily the views of the Department of Veterans Affairs. This material is the result of work supported with resources and the use of facilities at the San Francisco Veterans Affairs Medical Center.

### Funding

The authors declare there was no funding for this work.

### Competing Interests

The authors declare they have no competing interests.

### Author Contributions

- Pierre-Cédric B. Crouch conceived and designed the experiments, performed the experiments, analyzed the data, contributed reagents/materials/analysis tools, wrote the paper, prepared figures and/or tables, reviewed drafts of the paper.
- Carol Dawson Rose and Susan L. Janson wrote the paper, reviewed drafts of the paper.
- Mallory Johnson contributed reagents/materials/analysis tools, wrote the paper, reviewed drafts of the paper.

### Human Ethics

The following information was supplied relating to ethical approvals (i.e., approving body and any reference numbers):

This study was approved by the Committees on Human Research at the University of California, San Francisco. Approval number 12-08729.

### Supplemental Information

Supplemental information for this article can be found online at http://dx.doi.org/10.7717/peerj.852#supplemental-information.

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
