# Peer review of "A pilot study to evaluate the magnitude of association of the use of electronic personal health records with patient activation and empowerment in HIV-infected veterans"

_PeerJ, doi:10.7717/peerj.852_

## Round 0.1 · original submission · Minor Revisions

The reviewers both felt that this is an interesting and worthwhile paper. Their comments were appropriate and well-considered. Addressing these concerns would make this a much stronger paper.

·

Basic reporting

The article meets basic reporting requirements.

Experimental design

This is an observational cross-sectional study of 40 patients with HIV who completed a survey and whose medical records were reviewed. The research question is defined, relevant and meaningful. It is whether patient who use an personal health record (PHR) have greater health empowerment, patient activation, and satisfaction with health care compared to patients not using a PHR.

This design is not particularly strong in terms of assessing or inferring any kind of causality (as the authors note in the limitations section); however since there is relatively little research in this area with HIV positive patients it is worthwhile.

It was noted that the patients were recruited from the San Francisco VA Medical Center, Infectious Disease Clinic, and yet the human subjects approval was from the University of California, San Francisco. The authors should clarify why no human subjects approval from the San Francisco VA medical center was needed.

Validity of the findings

Validity of the findings:

The study was well conducted. However there are some things that could improve it.
1. It was unclear why both Patient Activation (PAM) and Patient Empowerment were measured. They seem to be very similar measures. Please explain how they differ and why both were selected.
2. Education was analyzed with a seven category variable. Given the small sample size (n=40; with 20 in each group), it may not be surprising that there was no statistical difference. I wonder if the variable were dichotomized (4 year college or more vs. 2 year college or less) if the there would be a statistically significant (or near significant) result. I suggest the authors perform that test and report it. There is the appearance of a more education in the group of patients who use the PHR
3. The first sentence of the 3rd discussion paragraph reads, “…our findings suggest that the use of electronic personal health records provides patients with more knowledge…”. This feels like it goes beyond reasonable conclusion, and implies causality, which the authors themselves note later on cannot be inferred from these data, with this type of study design. Consider saying, “We speculate…; or “We hypothesize that…”
4. Please explain what the question was that asked about access to computer. This is important because it is unclear if the question was about having a computer at home, or owning a computer, vs. the more broad question of having the ability to use a computer when one wants to or needs to (e.g. using a friend’s, using one for free at the library, etc.).
5. The measure of adherence was self report. This is one way of assessing, but since the study team had access to the medical records it would seem that also measuring adherence using some kind of pharmacy refill mechanism (assuming the patients were getting their ARV medications from the Veterans hospital) would have been another way to assess adherence without the social bias that comes in to play with self report.

Additional comments

This is useful study to help the field continue to make progress in understanding what role PHRs can play in aiding patients to manage chronic conditions.

The major limitation of this study is that there is a high probability of confounding by indication. And the socio-demographic data underscore this, with differences by race and probably by education – i.e. the differences in empowerment, viral load, etc. are more likely to be due to psycho-social and socio-demographic factors than they are to be due to the use of the PHR. The authors acknowledge this and provide recommendations for future studies that would help better determine what role if any PHRs play in the health, empowerment, etc. of patients.

The introduction has a sentence that seems to imply that the San Francisco VA developed the MyHealtheVet ePHR and that it is a system only in use at SFVAMC. I believe it was more of a national VA-wide effort; and it is a system available to Veterans in every VA medical system.

The literature cited in the discussion covers a number of studies related to PHR use. It would be helpful to break them down somewhat into those studies in which “passive PHR use” was examined (such as the current study) vs. those studies in which there was an intervention which used PHR as part of the intervention delivery. I think the Green, Cook and Ralston article may fall into that latter category. My impression is that studies in which PHR is part of an intervention have shown more promising results thatn studies of passive PHR use.

The Discussion indicates the authors believe this is the first study of this kind. This reviewer has been involved in two that have information about HIV patients’ use of PHRs – one was observational using survey and medical record data, one was interventional. The former had findings that are very much in line with the current study.

The full name of CAHPS may not be completely correct. I believe P may be for Providers and S for Systems.

·

Basic reporting

The article is well written, provides important context via literaure citations, and is coherent. Additional suggestions to improve are included in the comments to the authro.

Experimental design

The methods are appropriate and use quantitative techniques to explore measures of interest with data collected utilizing established and validated instruments. Suggest additional review by statistician may be needed to validate (this is not my area of expertise).

Validity of the findings

The authors contextualize the findings appropriately and identifies validity and also scope (associations but not causal influences). The conclusions are appropriate and based on the data. Study limitations are numerous but the authors are careful to be explicit about these.

Additional comments

A pilot study to evaluate the magnitude of association of patient activation in HIV-infected veterans who use electronic personal health records

General comments
=============
This paper describes patient characteristics and patient activation, empowerment, satisfaction, knowledge of their CD4, Viral Loads, and antiretroviral medication, and medication adherence outcomes associated with electronic personal health record use in Veterans living with HIV at the San Francisco VA Medical Center. This is an important topic that helps to address some of the current gaps in the PHR literature.

Specific comments
=============
Major comments
* * *
1. The methods are appropriate and use quantitative techniques to explore measures of interest with data collected utilizing established and validated instruments.
2. As a pilot study, there are some limitations, however the authors describe this explicitly and identify areas for further research.
3. The reference to Meaningful Use criteria is important and description of explicit MU criteria is helpful.
4. Although the study was conducted at a particular site please clarify that the My HealtheVet patient portal is a nation-wide system. It may be useful to include an additional citation or two and also to add a description of the available features for contextualizing this ePHR. The brief reference in Table 2 may be insufficient and “email” should be changed to “secure messaging”.

For the authors’ consideration:

• Nazi KM, Hogan TP, Wagner TH, McInnes DK, Smith BM, Haggstrom D, Chumbler NR, Gifford AL, Charters KG, Saleem JJ, Weingardt KR, Fischetti LF, Weaver FM. Embracing a health services research perspective on personal health records: lessons learned from the VA My HealtheVet system. J Gen Intern Med. 2010 Jan;25 Suppl 1:62-7. Review. PubMed PMID: 20077154; PubMed Central PMCID: PMC2806958.
• Nazi KM, Turvey CL, Klein DM, Hogan TP, Woods SS. VA OpenNotes: exploring the experiences of early patient adopters with access to clinical notes. J Am Med Inform Assoc 2014;0:1–7.
• McInnes DK, Solomon JL, Shimada SL, Petrakis BA, Bokhour BG, Asch SM, Nazi KM, Houston TK, Gifford AL. Development and Evaluation of an Internet and Personal Health Record Training Program for Low Income Patients with HIV and Hepatitis C. Medical Care, 2013 Mar:51 Suppl: S62-S66. doi: 10.1097/MLR.0b013e31827808bf

5. Please clarify if the VA IRB also approved the study (page 7 notes: The Committees on Human Research at the University of California, San Francisco, approved this study (CHR# 12-08729).
6. I’m not sure they being able to phonetically spell the ART is an adequate measure of knowledge of what the treatment plan entails, and indeed only 45% were able to do so. More importantly, 93% reported full adherence. Suggest that clarification on this be attempted since I would think it more important that patients be able to describe the treatment plan then to be able to identify what could likely be somewhat of a cryptic name—with no bearing on understanding the plan of care and patient actions needed for concordance with the plan.
7. In Table 1, gender does not add up correctly to the sample size for My HealtheVet users. Please also clarify the calculation of male to female ration. If all participants were male, how was this calculated and what value does it add?
8. It may important to also mention in describing the sample the presence of co-morbid conditions in both samples.
9. Suggest additional review by statistician may be needed to validate (this is not my area of expertise).
10. Overall this paper is an important contribution to the literature.

Minor comments
* * *
1. Please correct all references to “MyHealtheVet” to read “My HealtheVet” and correct the web address on page to: http://www.myhealth.va.gov

---

## Round 0.2 · Minor Revisions

Although both reviewers agreed that this revision represented a significant improvement that addressed concerns from the first round of reviews, they also identified several minor issues that can easily be addressed. Please make these revisions.

·

Basic reporting

This is a re-review. This meets standards for basic reporting.

Experimental design

The article meets standards for experimental design.

Validity of the findings

The article meets standards for validity of findings.

Additional comments

General Comments for the Author
1. The authors have addressed reviewer comments.
2. The authors might consider revising the title. It is unclear currently which association is being examined. I think something like “…association of use of electronic personal health records with patient activation in HIV infected veterans…” might make that clearer.
3. Also, regarding the title, heatlhcare empowerment seems to also be an important part of the study objective, so why not include that in the title too? (and it is a part of the theoretical model behind the research)
4. Small typo in the “Theory” section: should be “The variables collected in …
5. Methods: The sample is called a “Non-probability quota sample”. I realize that there is some accuracy to this, but after reading the rest of the description I would prefer to see this described as a “convenience sample” (with the details that you have that indicate the 20 and 20 part).
6. Measures section: the authors indicate in reference to CAHPS: “These composite scores have low reliability with a Cronbach’s alpha ranging from .58-.75…”. I suggest dropping the words “have low reliability”. While there is some debate over cutoffs, I think many researchers would consider .75 to be in the fair to good range. How about just saying “These composite scores have Cronbach’s alphas ranging from…”
7. Analysis: there is mention of univariate analysis. Since the table is examining one variable (e.g. age) stratified by another (MHV use) this would seem to be a bivariate analysis.
8. In results, the authors have added a sentence (just before the discussion) that describes what is meant by access to a computer. This probably is better suited for the methods section since it is the definition of one of the measures.
9. Discussion. I think it would help the reader if studies could be grouped by those that were observational studies of PHR use, versus those that were prospective controlled studies in which the PHR was part of an intervention (such as the Green article). Unless that is made clear there is an implicit comparison of quite different approaches to PHR use.
10. Discussion, last paragraph. The first sentence implies that Wagner found that patients benefitted from PHR use in terms of controlling their hypertension. This was a randomized trial and the main finding was that the PHR did not help hypertension control, “No impact of the PHR was observed on blood pressure, patient activation, patient perceived quality, or medical utilization in the intention-to-treat analysis.” I think readers will be misled by the way this is currently written. If the authors want to refer to the sub-analyses in which greater frequency of use of PHR was associated with greater control, they should make it clear that it was a secondary finding and they should indicate that the primary finding was that PHR did not help with hypertension control.

·

Basic reporting

The article is well written, provides important context via literature citations, and is coherent. In this revisions, corrections have been made to correct My HealtheVet spelling and context as requested.

Experimental design

The methods are appropriate and use quantitative techniques to explore measures of interest with data collected utilizing established and validated instruments.

Validity of the findings

The authors contextualize the findings appropriately and identifies validity and also scope (associations but not causal influences). The conclusions are appropriate and based on the data. Study limitations are numerous but the authors are careful to be explicit about these.

Additional comments

Review of Revision 012615

1. On page 4 the reference to ‘Veterans Affairs Medical Center’ should be changed to ‘Department of Veterans Affairs’ when referring to the launch of My HealtheVet since the portal is a national system.

The Department of Veterans Affairs has been a leader of ePHR development with its system named My HealtheVet, which launched in 2003.(United States Department of Veterans Affairs, 2013) The My HealtheVet ePHR (http://www.myhealth.va.gov) is a tethered system that allows all Veteran patients to have full access to their lab results, progress notes, and medication refills. It also gives patients the ability to send secure messages to their providers. Patients are required to have their identity authenticated in person before obtaining access.

2. To clarify the gender portion of the sample could you note that one participant was transgender?

---

## Round 0.3 · accepted · Accept

Thank you for your careful attention to the comments of the reviewers.